# The Influence of Constraints on Gelation in a Controlling/Living Copolymerization Process

**DOI:** 10.3390/ijms24032701

**Published:** 2023-01-31

**Authors:** Piotr Polanowski, Andrzej Sikorski

**Affiliations:** 1Department of Molecular Physics, Łódź University of Technology, Żeromskiego 116, 90-924 Łódź, Poland; 2Faculty of Chemistry, University of Warsaw, Pasteura 1, 02-093 Warsaw, Poland

**Keywords:** confined space, Dynamic Lattice Liquid model, gelation, Monte-Carlo simulation, polymerization

## Abstract

We developed a simple model of the copolymerization process in the formation of crosslinked macromolecular systems. A living copolymerization was carried out for free chains, in bulk and in a slit, as well as for grafted chains in a slit. In addition, polymer 2D brushes were placed in a slit with initiator molecules attached to one of the confining walls. Coarse-grained chains were embedded in the vertices of a face-centered cubic lattice with the excluded volume interactions. The simulations of the copolymerization processes were performed using the Dynamic Lattice Liquid algorithm, a version of the Monte Carlo method. The influence of the initial initiator to cross-linker ratio, slit width and grafting on the polymerization and on the gelation was examined. It was also shown that the influence of a confining slit was rather small, while the grafting of chains affected the location of the gel pint significantly.

## 1. Introduction

Materials such as polymer networks and gels are now commonly used in a variety of industries, from automotive manufacturing to drug delivery, artificial muscles and stretchable electronics. For these types of materials, one of the main problems from the point of view, both in theory and in industrial application, is determining the gel point, the point at which the reacting liquid turns into a solid.

The cross-linking of polymers changes the properties of macromolecular systems [1] and eventually causes gelation. An increase in the number of cross-links leads to the formation of insoluble gel. The transition from sol to gel is called the gel point [2]. Polymer gels can be used as functional materials in technology and biomedicine [3,4]. Gels can be prepared from monomers and cross-linkers using the controlled radical polymerization method (CRP) [5,6,7,8,9]. The process of free radical polymerization was recently reviewed [10]. Theoretical studies on gelation were pioneered by Flory and Stockmayer (F–S theory) [11,12] for the case of free radical polymerization; however, it has been shown that the gel point was observed at considerably higher monomer concentrations in both experiments [5,8,9] and simulations [6,13,14,15].

Confinement, i.e., geometric constraints, changes the structure and physical properties of macromolecular systems [16]. The gelation process, under confinement, was studied experimentally, revealing differences between the structure and properties of a gel obtained in such a way and one in bulk [17]. Polymer brushes, i.e., macromolecular structures with chains densely tethered onto a surface, recently became the subject of many experimental works because of their practical importance [18,19,20,21,22]. They have also been studied using Molecular Dynamics and Dissipative Particle Dynamics [23,24,25], Monte Carlo simulations [26,27,28,29,30,31,32,33], scaling theory and self-consistent field theories [34,35,36,37,38]. A small number of Monte Carlo simulations were devoted to cross-linked grated chains (polymer brushes) [39,40]. Cross-linked polymer brushes are considered to be good drug carriers [39] and therefore, have already been obtained in experiments [40,41,42,43] and studied using computer simulations [44].

In this work, the controlled/living copolymerization process of a monovinyl and a divinyl monomer were studied for systems with some geometrical constraints. Copolymerization was performed in bulk, in a slit and for chains grafted onto one wall of the slit. Thus, in the latter case, we obtained brushes synthesized by a grafting-from procedure, in which the polymerization of the chains started from an initiator attached firmly to a surface. The influence of the initial initiator to cross-linker ratio on the polymerization parameters was the first objective of this study. The second objective was to study the influence of the constraints used (slit and grafting) on the copolymerization and gelation processes. Because of the complex architecture, large size and high density, the macromolecular systems were studied employing a coarse-grained lattice model. During and after polymerization, the system contained flexible chains immersed in monomer (a good solvent); monomer and cross-linker molecules were explicitly included in the model. The model, called the Dynamic Lattice Liquid (DLL), was used. It was a cooperative Monte Carlo simulation method based on the concept of cooperative movement with rearrangements of objects in the form of closed loops of displacements [45]. The main advantage of this model was the possibility of studying macromolecular systems at high densities. This algorithm was successfully used previously in studies of various polymerization processes, including gels [5,6,13,14,46,47], polymer brushes [31,48], star-branched polymers [46,49,50], hyperbranched polymers [48], dendrimers [49,50] and shape-memory polyurethanes [51]. The influence of constraints on the structure of cross-linked polymers will be discussed in our next paper.

This paper has been organized, as follows. In *Methods and Materials* section, we introduced the model of polymer systems and the details of the simulation method called Dynamic Lattice Liquid. The *Results and Discussion* section was divided into two subsections; the first (*The Polymerization Process*) containing a study of polymerization in which basic parameters of this process were described and discussed, and the second (*Location of the Gel Point*) focusing on the problem of determining the gel point and its dependence on imposed geometrical constraints. In the last section, *Conclusions*, we summarized the results and discussed the most important conclusions.

## 2. Results and Discussion

### 2.1. The Polymerization Process

In order to characterize the polymerization process in the systems studied, we used commonly used quantities describing this process, as follows:

number-average degree of polymerization of all macromolecules *P_n_*(*α*) as a function of conversion
(1)$$P_{n}\left(\alpha \right)=\sum_{i=1}n_{i}p\left(\alpha ,n_{i}\right)$$
where α is the conversion of the monomer (*conv_M_*) or of the cross-linker (*conv_X_*), *n_i_* represents the chain length of each polymer population (i.e., monomer is not taken into account), *p*(*α,n_i_*) represents the fraction of molecules of chain length *n_i_* and $\sum_{i}p\left(\alpha ,n_{i}\right)=1$ for each *α*;weight-average degree of polymerization *P_w_*(*α*) as function of conversion
(2)$$P_{w}\left(\alpha \right)=\frac{\sum_{i=1}n_{i}{}^{2}p\left(\alpha ,n_{i}\right)}{\sum_{i=1}n_{i}p\left(\alpha ,n_{i}\right)}$$dispersity, which is defined as $\frac{P_{w}\left(\alpha \right)}{P_{n}\left(\alpha \right)}$.

Figure 1 shows simulation results of bulk copolymerization (system A) of a vinyl monomer and divinyl cross-linker for various [Ini]_0_/[X]_0_/[M]_0_ ratios. *P_n_*, *P_w_* and *P_w_*/*P_n_* were plotted as a function of monomer conversion. The plots of *P_n_* and *P_w_* were plotted in a semi-logarithmic scale to enable studies of the chain mass at both low and high monomer conversion. We chose monomer conversion as the independent variable because precise definitions of time in Monte Carlo simulations are unclear. One can observe a typical evolution of the degrees of polymerization *P_n_* and *P_w_* as a function of the monomer conversions *conv_M_*. Thus, in this case, there was a rapid increase of *P_n_*, *P_w_* and a characteristic curve of dispersity for cross-linking process [2,46]. The location of the rapid increase in *P_w_* depended strongly on [Ini]/[X]. For higher values of monomer conversion, the average chain lengths lay along straight lines, indicating that the increase of *P_n_* was approximately exponential. The higher [Ini]/[X] ratio, i.e., the higher concentration of the crosslinker, led to a considerably higher degree of polymerization of *P_n_* than expected [46].

The above behavior of *P_n_*, the weight-average *P_w_* and *P_w_*/*P_n_* could now be compared with the results obtained when the polymerized system was subjected to geometric constraints, i.e., with system B, where the chains were inserted into a slit built of parallel walls with a spacing of *d* = 100 and 10 lattice units. In the situations considered, the ratio [Ini]/[X] in the system was maintained as before. By changing only the volume, the amount of monomer was changed. Figure 2 presents the comparison of the simulation results of the system II (polymerization in a slit) with *P_n_*, *P_w_* and *P_w_*/*P_n_* shown as a function of monomer for various [Ini]_0_/[X]_0_/[M]_0_ ratios. As shown, it was observed that, for a wide slit (the distance between the walls is *d* = 100) the simulation proceeded in a similar way to the system I (without geometric constraints). A similar behavior of the curves in the corresponding cases was observed. In the case of the narrow slit, where the distance between the walls was drastically reduced to *d* = 10, we observed a weaker increase in both degrees of polymerization (*P_w_* and *P_n_*) as the polymerization process progressed. This could have been related to a much smaller amount of monomer, but the nature of these changes, i.e., the characteristic points where rapid changes occurred (which were important for gelation), were not significantly different from the two previously considered cases (system I and a wide slit in system II). The degree of polymerization *P_n_* was an order of magnitude lower than that for a free solution, regardless of the value of the [Ini]/[X] ratio. Therefore, the main factor affecting the evolution of the system was the ratio of the initial cross-linker concentration to the initial initiator concentration, while the amount of monomer did not matter. It was also observed that the shapes of the dispersity curves were similar in all three cases considered. The maximum values of dispersity in a wide slit were comparable to those for bulk—while, in a narrow slit, they were slightly lower.

In the last system considered (system C), the cross-linker was randomly distributed in a slit, while the initiator was located on one wall of the slit. The grafting density *σ* (in this case, a fraction of lattice sites occupied by the initiator to the total number of lattice sites on the wall) changed between 0.05 and 0.60. For a system in which polymers were grafted to one of the walls, the polymerization process is shown in Figure 3. In the case of system C, in addition to the geometric constraint (in the form of parallel walls), the polymers were grafted onto one of them. We observed that the polymerization process was different from the cases of systems A and B, described earlier. All tested parameters behaved differently than in the previously described cases, i.e., the values *P_w_* and *P_n_* were significantly lower during the entire polymerization process, while the dispersity *P_w_*/*P_n_* reached higher values. Moreover, compared to systems A and B, the characteristic points where a rapid increase occurred shifted to lower values. The differences were especially visible for a wider slit (*d* = 100) where the dispersity did not decrease at higher monomer conversions and remained high until the very end of the polymerization process. This could have been related, not only to the change in monomer amount, which naturally led to a slower increase in chains’ size, but also to the characteristic points (points where a rapid increase in the measured values began) which shifted, relative to each other, in the right and left panels.

The weight increase of the largest molecule in the system, along with the amount of the reacted monomer, was a behavior deemed important, as an addition to the information obtained from the above analysis. Figure 4 presents the weight of the largest chain (what corresponded to the longest length or the highest number of polymer beads) *n_max_* as a function of the monomer conversion *conv_M_* for all systems considered (systems A-B-C). From this Figure, it can be observed that the weight of the longest chains behaved similarly in all systems under consideration—although, in system C, the shape of the curves differed slightly. On the other hand, rapid growth of the largest molecule was observed at much lower monomer conversion for system C (where polymer chains, including the largest one, were grafted onto the wall), as compared to systems A and B (free solution and a slit, respectively). In the two latter cases, the increases in *n_max_* were very similar. These differences suggested a significant effect of grafting on the position of the gel point, and thus, as highlighted in the next subsection, we performed a more detailed investigation of the gelation process.

### 2.2. The Location of the Gel Point

To determine the gel point in the considered systems, we used a parameter called the reduced degree of polymerization (*RDP*), which was equal to the weight-average degree of polymerization without the longest chain in the system. It was defined analogously to *P_w_*(*α*):(3)$$RDP\left(\alpha \right)=\frac{\sum_{i}n_{i}^{2}p\left(\alpha ,n_{i}\right)-n_{max}^{2}}{\sum_{i}n_{i}p\left(\alpha ,n_{i}\right)}$$
where *n_max_* denotes the chain length of the longest macromolecule in the system (formed by the highest number of beads). This parameter, which has also been denoted as reduced average cluster size, proved to be very useful in the analysis of gelation processes, and was proposed by Hoshen and Koppelman [52]. The gel point was defined as the monomer and cross-linker conversions, respectively, at which the *RDP(α)* reached the maximum value. This procedure has been used for the detection of the gel point in simulations of the copolymerization ATRP process [10,14]. Figure 5 shows an exemplary application of *RDP* to determine the gel point, i.e., the monomer conversion rate in Figure 5a and the cross-linker conversion rate in Figure 5b at this point, both determined for a grafting density of 0.3 and a wall spacing *d* = 50. It can be observed that the sharp maxima observed in these plots were good indications of gel points.

Having the procedure of determining the gel point established, we calculated this parameter for system B (polymerization in a slit) for some values of the density of initiator (0.05, 0.10 0.30 and 0.60, (i.e., a different number of chains)) and for some widths of the slit between a wide and a narrow case (*D* = 10, 20, 50 and 100). Figure 6 presents the monomer conversion at the gel point as a function of the ratio of initiator and cross-linker concentration [Ini]/[X] for the mentioned above concentrations of the initiator and slit widths. Monomer conversions *conv_M_* at the gel point, determined by the same method for the free solution case (system A), were included for comparison. As observed, geometrical constraints did not affect the location of the gel point; however, it is important to remember that the narrower slit corresponded to a smaller amount of the monomer. The behavior of the gel point in a slit was very close to that in a free solution, and the differences decreased as the width of the slit decreased. This decrease was almost linear on the plot, implying a rough power dependence.

The location of the gel point could also be studied, depending on the cross-linker conversion. Figure 7 presents the cross-linker conversion *conv_X_* at the gel point as a function of the ratio of initiator and cross-linker concentration [Ini]/[X] for some densities of initiator and some slit widths (system B). The values of the initiator density and width of the slit were the same, as in the case of the monomer conversion discussed earlier (Figure 6). The cross-linker conversions at the gel point, determined for the free solution case (system I), were included for comparison. The behavior of the gel point was not significantly different from that of the monomer (Figure 6); the dependence of the gel point on both parameters was rather weak and the plots were almost linear, suggesting a rough power dependence. The range of the gel point changes was similar to that of a free solution case (system A).

Analogical calculations concerning the location of the gel point were carried out for chains grafted in a slit (system C). Figure 8 presents the monomer and cross-linker conversion at the gel point for grafted chains as a function of the ratio of initiator and cross-linker concentration [Ini]/[X] for the same grafting densities as for systems A and B. Presented values concerned a narrow slit with *D* = 10. The monomer and cross-linker conversions at the gel point determined for the free solution case (system A) were included for comparison. The behavior of the gel point was not significantly different from those of the free solution and the slit presented in Figure 6 and Figure 7 and discussed above. All plots exhibited the same behavior as in systems A and B, i.e., the decrease of the gel point location was almost linear. There was only one exception found for the case of the largest grafting density (*σ* = 0.60) where the gel point was reached at considerably lower monomer and cross-linker conversion, especially for intermediate and higher values of the [Ini]/[X] ratio.

The next question concerned the influence of the width of the slit on the location of the gel point. For this purpose, we analyzed the gel point for different grafting densities in a wide slit (*D* = 100). Figure 9 presents the monomer and cross-linker conversion at the gel point for grafted chains (system C) as a function of the ratio of initiator and cross-linker concentration [Ini]/[X] for some grafting densities (*σ* = 0.05, 0.10, 0.30, and 0.60). The monomer and cross-linker conversion at the gel point determined for the case of a free solution (system A) were included for comparison. The plots were almost linear and we observed that the gel point was reached at lower monomer and cross-linker concentrations than in the narrow slit (with *D* = 10, Figure 8). Contrary to the previous case, the dependencies of the gel point on [Ini]/[X] on the grafting density were definitely stronger.

Based on the gel point location discussed above, a question arose whether both the width and density of a monomer layer in the slit had effects on the location of the gel point. For this purpose, we analyzed the gel point for different slit widths (*D* = 10, 20, 50 and 100) at the grafting density *σ* = 0.05, 0.10, 0.30 and 0.60). Figure 10 shows the monomer conversion at the gel point for the polymerization of grafted chains (system C) in a slit as a function of the ratio [Ini]/[X] for a given grafting density while varying the slit width. Monomer conversions at gel points determined for the free solution case (system A) were included for comparison. We observed that the deviation of the monomer conversions at the gel points from those in a free solution increased with increases of the grafting density and slit width. At relatively low grafting densities, significant differences were only present for wider slits.

An analysis analogous to the previous one was performed, i.e., studying the location of the gel point determined from the cross-linker conversion. Figure 11 presents the cross-linker conversion at the gel point for the polymerization of grafted chains in a slit (system C) as a function of the ratio [Ini]/[X] for different grafting densities and slit widths. Cross-linker conversions at the gel point determined for the free solution case (system A) were included for comparison. The conversion curves were very close to the previous case (Figure 10). We observed that the deviation from the cross-linker conversions at the gel points (compared to those in a free solution) increased with increases of the grafting density and slit width. As shown in Figure 10 and Figure 11, in order to lower the gel point, it was necessary to use less confinement (a wider slit in our case) and a higher grafting density.

## 3. Methods and Materials

In the simulations presented, a coarse-grained approach was used. In the Dynamic Lattice Model (DLL), details of the chemical structure of monomers and polymer molecules were disregarded. A unit (monomer/mer built of many atoms) was considered an elementary structure. Its status changed by polymerization and each reaction step resulted in the formation of an unbreakable chemical bond which reduced the number of molecules in the system by one. The reactivity of all functional groups (the probability of being selected for a reaction) was constant, with no substitution effects being considered, and independent of the chain length. The probabilities of all the reactions were assumed to be 0.02, based on previous polymerization studies [5,13]. In most cases, no termination or chain transfer reactions were considered and thus, it was an ideal living copolymerization. This assumption is not strictly true for controlled/living radical polymerization, such as Atom Transfer Radical Polymerization (ATRP), but should be a reasonable approximation in this case [5,13]. The polymerization reaction was simulated according to the general scheme, which was developed and presented elsewhere in detail [5,53]. Polymerization was carried out until the monomer was completely depleted. The model used can be treated as Monovinyl monomer and Divinyl cross-linker copolymerization [5].

The DLL model has been described in more detail in previous publications [13,54,55,56] so its main features are only briefly summarized here. It is a lattice model in which the beads, representing the monomers, occupy all the lattice sites of the system (density factor *ρ* = 1). A field of randomly chosen unit vectors represents motion attempts. These vectors are assigned to the objects in the system and point in the directions of the attempted motions. Only those attempts that coincide in such a way that the sum of displacements along a path involving more than two objects is equal to zero (continuity condition). This results in displacements of objects along self–avoiding closed loops, each bead to a neighboring lattice site. All objects which do not contribute to such correlated loops remain at their previous positions. 

We considered an athermal system, so all allowable rearrangements were performed. Molecular systems treated in this way can be considered to have been provided with a dynamic, consisting of local vibrations and occasional diffusion steps, resulting from the coincidence of attempts of the neighboring elements to displace beyond the occupied positions. Within a longer interval of time, this kind of dynamic can lead to displacements of individual objects along the random walk trajectories, with steps distributed randomly in time.

Simulations of cross-linking and gelation in the infinite volume (periodic boundary conditions were set in all directions) for some [X]_0_/[Ini]_0_ ratios were made for the sake of comparison (system A most left). Simulations of cross-linking and gelation in the infinite volume (no geometric constraints, periodic boundary conditions are imposed in all directions) were also performed for the sake of comparison (Figure 12, left). Periodic boundary conditions were assumed in the *x* and *y* directions, while solid, impenetrable walls were assumed at *z* = 0 and *z* = *z_max_*, i.e., the copolymerization in slits of width *d* = *z_max_* was performed for the same values of [X]_0_/[Ini]_0_ ratio. The simulation box size was 100 × 100 lattice units in the *x*- and *y*-directions and varied in the *z*-direction with *z_max_* = 100, 50, 20 or 10 for a given simulation. For the simulations of brush growth in the slit, all initiators were attached to the wall *z* = 0 and cross-linkers were randomly distributed in the simulation box (Figure 12, right). In the case of simulations of free chains in the slit, the initiators and cross-linker were randomly placed in the simulation box (Figure 12, middle). All remaining lattice sites in the simulation box were occupied by the monomer. The number of initiators in a given simulation was based on the assumed grafting density *σ*, i.e., the ratio of the lattice sites occupied by the initiator to all available sites. The same number of initiator molecules per simulation box was set in the corresponding simulations of free chains. The number of cross-linkers decreased with the width of the slit to maintain constant density of these elements. The ratio of the initial cross-linker concentration to the initial initiator concentration ([X]_0_/[Ini]_0_) varied between 1 and 7 in different simulations. These values were equal to half of the average number of crosslinks per chain at full conversion (when inter- and intrachain cross-links are counted).

## 4. Conclusions

We carried out simulations of a simple model of living polymerization with some geometrical constraints by means of the Monte Carlo method. The Dynamic Lattice Liquid algorithm, based on the cooperative movement concept, was employed with realistic reaction parameters. In this model, cooperative rearrangements of a system had the form of closed loops of displacements on face-centered cubic lattice, with all lattice sites occupied by cross-linker, monomer and growing macromolecules. The main focuses of this work were the kinetics of the polymerization and the influence of the constraints on the gelation process. In the frame of our model, we were able to give a qualitative description of the kinetic behavior.

Polymerization in a slit formed by two impenetrable walls proceeded quite similarly to that in a free solution (without geometrical constraints). The gel point was found to be located at nearly the same degree of conversion of monomer and cross-linker in both cases. Changes in the width of the slit did not significantly affect the gel point, even if the width was one order of magnitude narrower. When the initiator was located on one of the walls, i.e., when chains in the slit were grafted, the gel point shifted toward lower monomer and cross-linker conversion. The effects of the grafting density were found to be similar, i.e., higher density led to a decrease in both conversions. The reason for these differences was the predominance of inter-cross-linking in the initial period, especially at high grafting densities; other chains were attached to a growing grafted chain more often than monomers. Therefore, under such conditions, the gel points were located at lower monomer and cross-linker conversions, compared to polymerization in a free solution or a slit. It should be noted that the influence of grafting on the gel point was considerably stronger compared to polymerization in a free solution with intra-cross-linking disabled [49]. At low grafting densities, polymerization proceeded similarly to that in a free solution, where the mechanism of intra-cross-linking predominated. This could be explained by larger distances between the growing chains in the initial period of synthesis.

We observed that the deviations of the cross-linker conversions at the gel points (from those in a free solution) increased with increases of the grafting density and slit width.

## Figures and Tables

**Figure 1 ijms-24-02701-f001:**
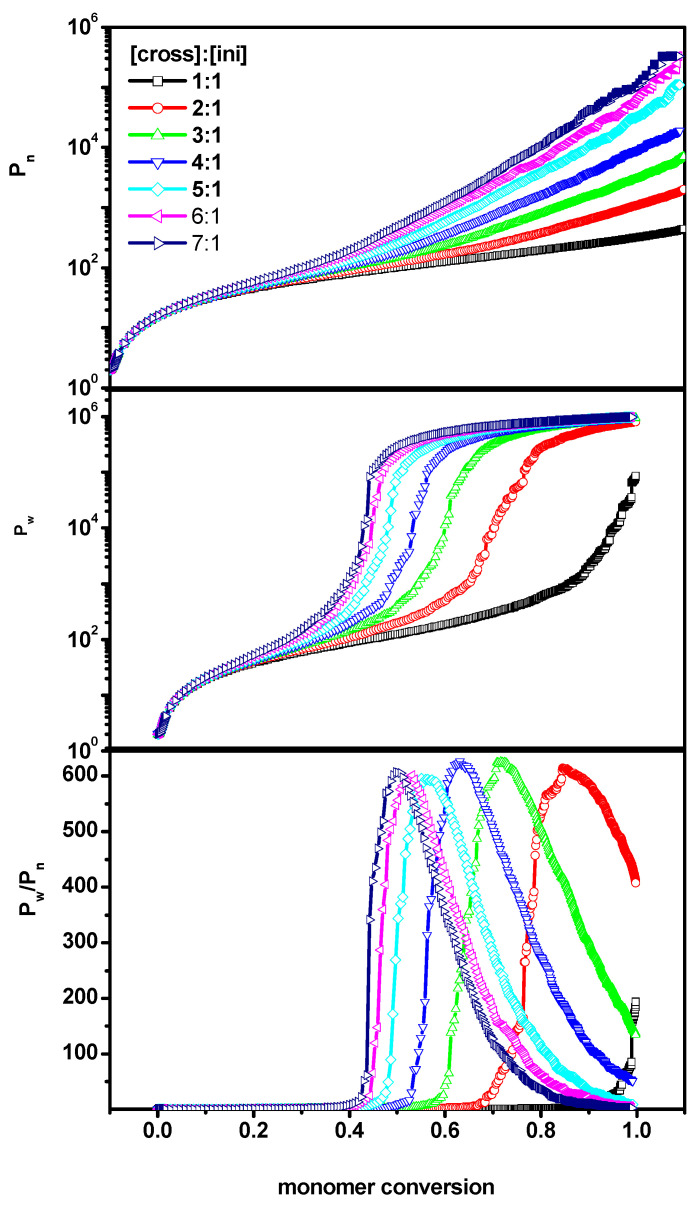
The number-average chain length (*P_n_*) (**upper**), the weight-average (*P_w_*) (**middle**) and the dispersity (*P_w_*/*P_n_*) (**lower**) vs. monomer conversion for the initiator concentration 0.6%. The case of a free solution (system A). The values of the ratios [Ini]/[X] are given in the inset to the upper Figure.

**Figure 2 ijms-24-02701-f002:**
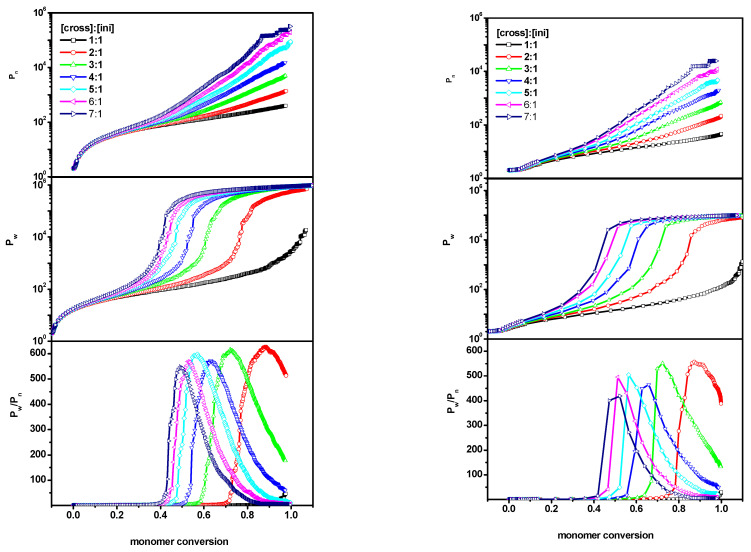
The number-average chain length (*P_n_*) (**upper**), the weight-average (*P_w_*) (**middle**) and the dispersity (*P_w_*/*P_n_*) (**lower**) vs. monomer conversion for the initiator concentration 0.6%. The case of the slit (system B) with the distance between walls *d* = 100 (**left** panel) and *d* = 10 (**right** panel). The values of the ratios [Ini]/[X] are given in the insets to the upper Figures.

**Figure 3 ijms-24-02701-f003:**
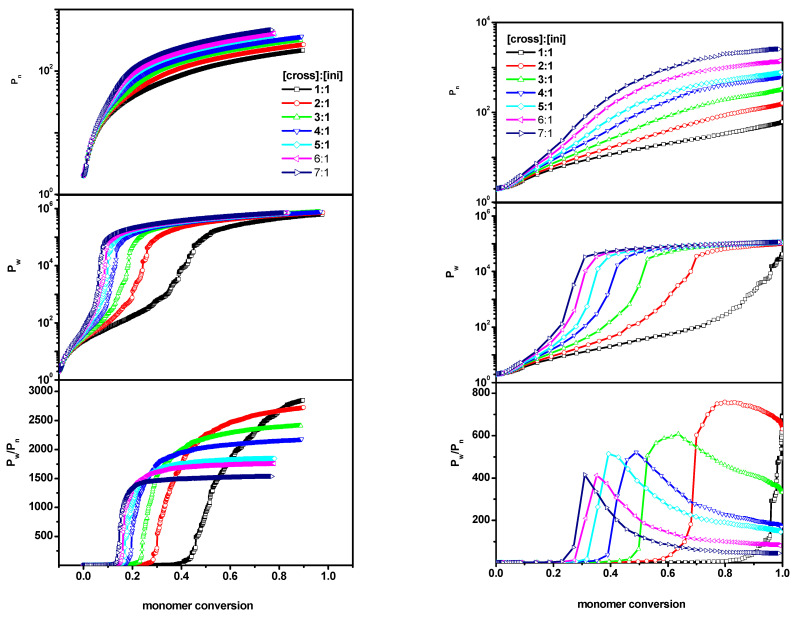
The number-average chain length (*P_n_*) (**upper**), the weight-average (*P_w_*) (**middle**) and the dispersity (*P_w_*/*P_n_*) (**lower**) vs. monomer conversion for the initiator concentration 0.6% (it corresponds to the grafting density *σ* = 0.6) in the slit with chains grafted to one of the walls (system C). The case of the distance between walls *d* = 100 (**left** panel) and *d* = 10 (**right** panel). The values of the ratios [Ini]/[X] are given in the insets to the upper Figures.

**Figure 4 ijms-24-02701-f004:**
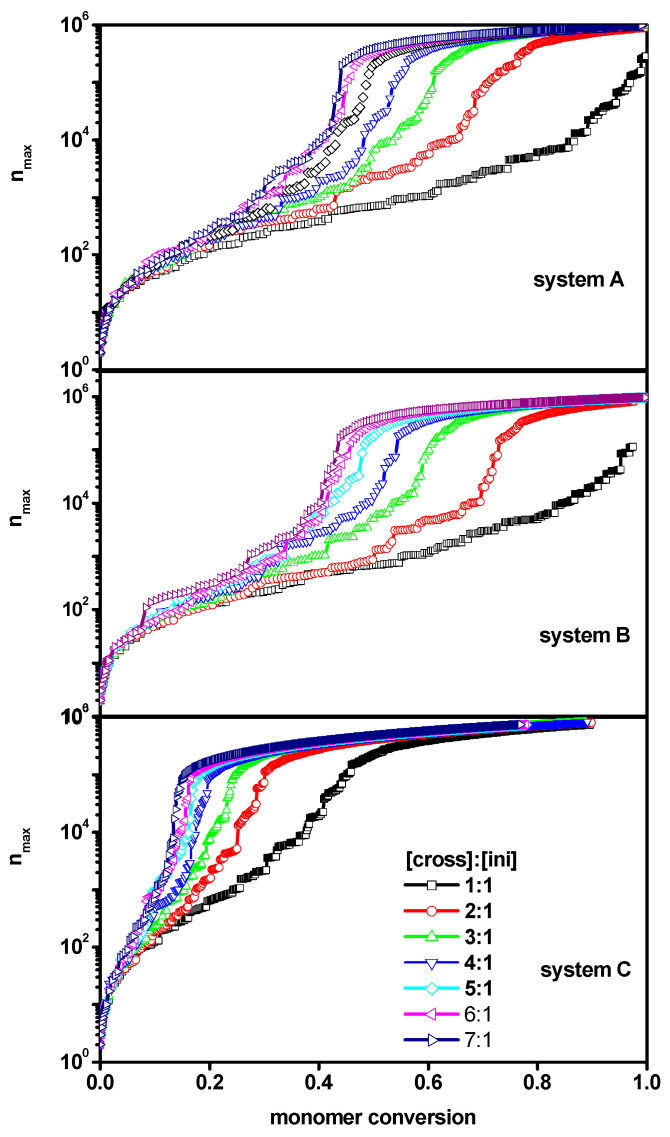
The evolution of maximum chain length with monomer conversion obtained for systems A, B and C. The initiator concentration is 0.6%, corresponding to the grafting density 0.6. The distance between the walls in the case of system B and C is *d* = 100. The values of the ratios [Ini]/[X] are given in the insets to the upper Figures.

**Figure 5 ijms-24-02701-f005:**
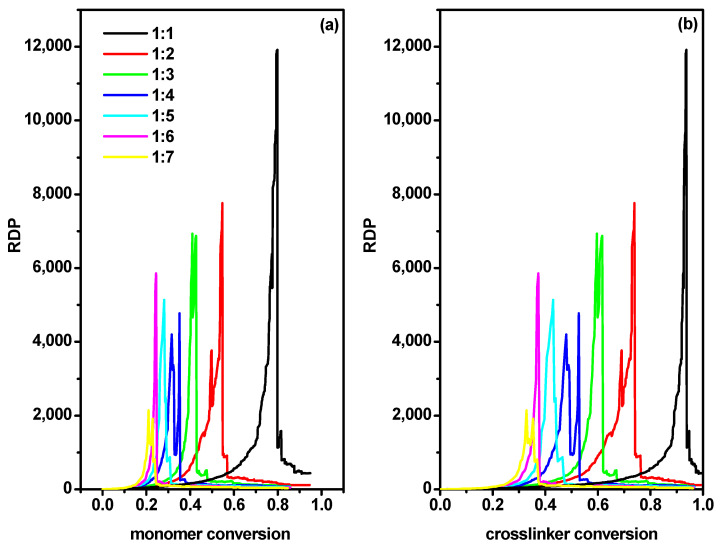
The reduced degree of polymerization as a function of monomer conversion (**a**) and the cross-linker conversion (**b**) for the distance between walls *d* = 50 and grafting density *σ* = 0.3. The values of the ratios [Ini]/[X] are given in the inset.

**Figure 6 ijms-24-02701-f006:**
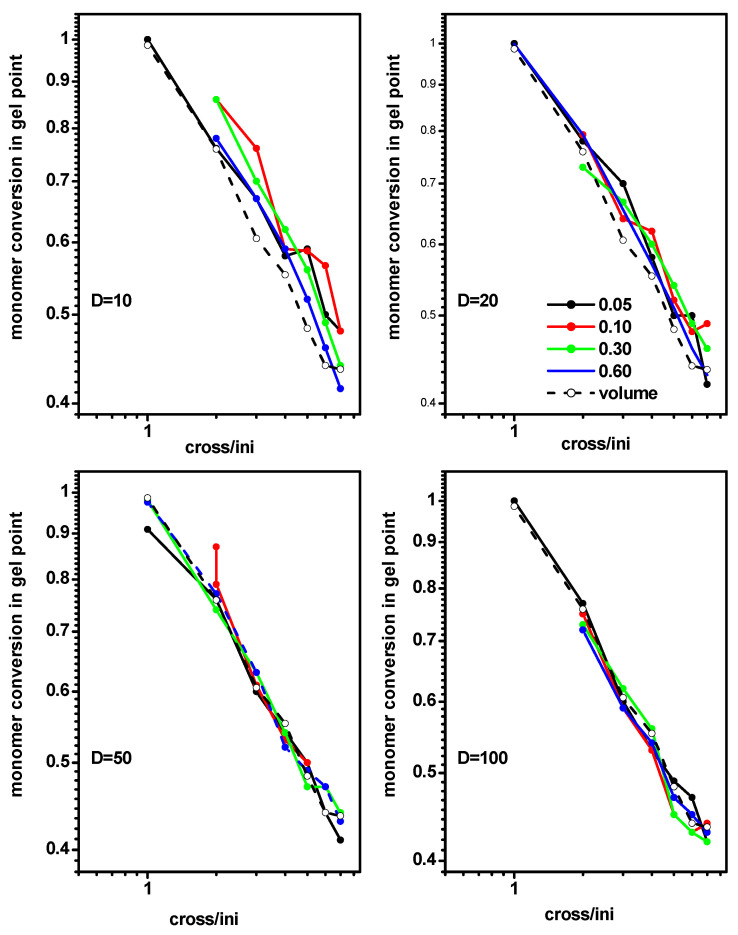
Monomer conversion at the gel point for the polymerization in a slit as a function of the ratio [Ini]/[X] for different widths of the slit. The initiator densities are given in the inset. The monomer conversion at gel point for a free volume is marked by the dashed line.

**Figure 7 ijms-24-02701-f007:**
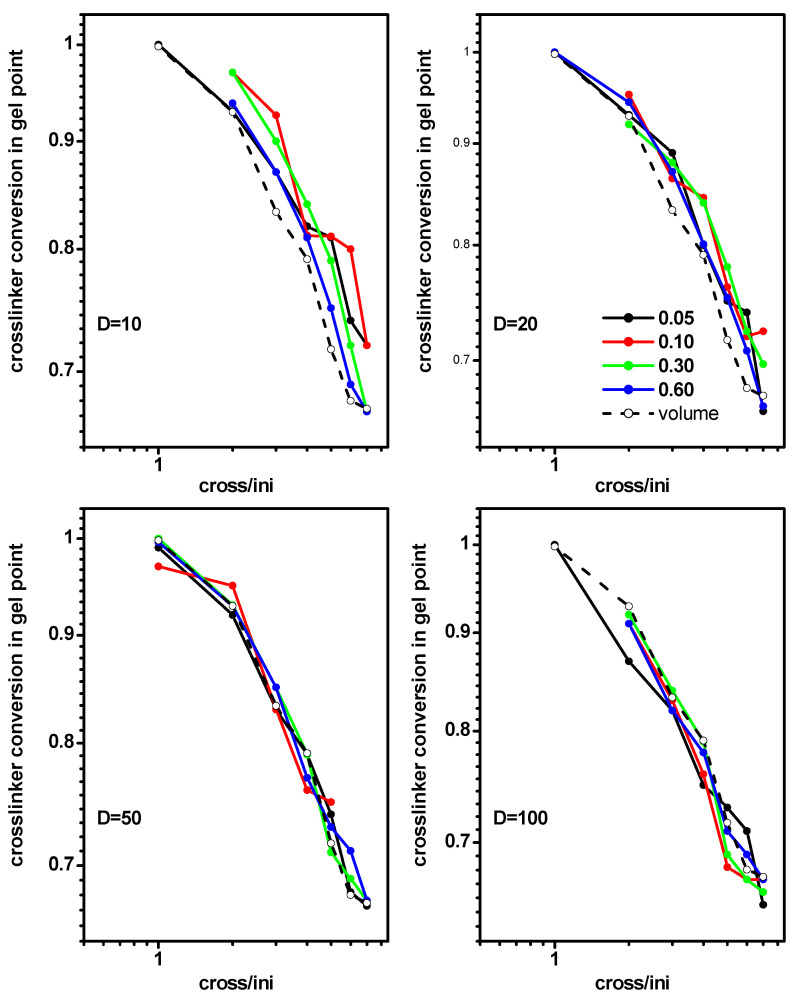
Cross-linker conversion at the gel point for polymerization in a slit, as a function of the ratio [Ini]/[X] for different widths of the slit. The initiator densities are given in the inset. The cross-linker conversion at gel point for a free volume is marked by the dashed line.

**Figure 8 ijms-24-02701-f008:**
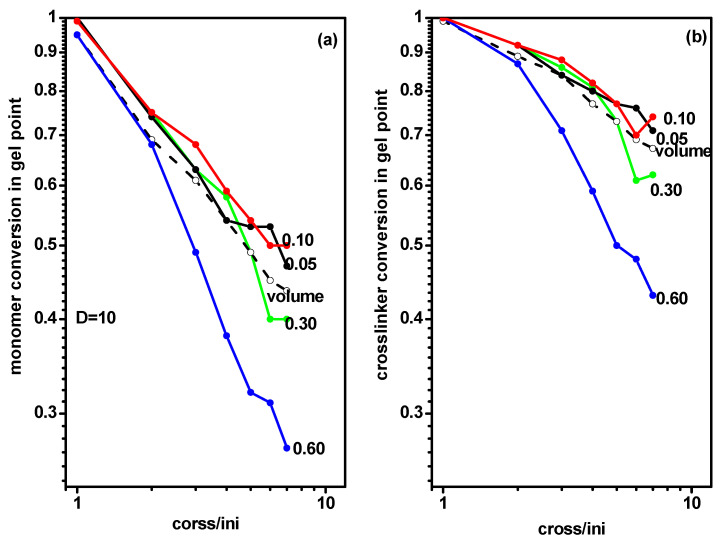
The conversion of monomer (**a**) and cross-linker (**b**) at the gel point during polymerization of grafted chains in a slit, with *d* = 10 as a function of the ratio [Ini]/[X] for different widths of the slit. The grafting densities are given in the inset. The conversion of monomer and cross-linker at the gel point for a free volume are marked by the dashed line.

**Figure 9 ijms-24-02701-f009:**
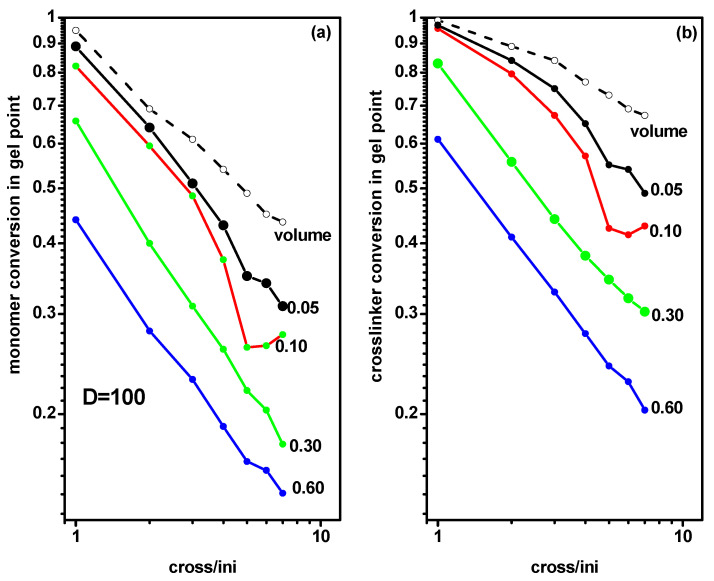
The conversion of monomer (**a**) and cross-linker (**b**) at the gel point for polymerization of grafted chains in a slit with *d* = 100 as a function of the ratio [Ini]/[X] for different widths of the slit. The grafting densities are given in the inset. The monomer and cross-linker conversion at gel point for a free volume are marked by the dashed line.

**Figure 10 ijms-24-02701-f010:**
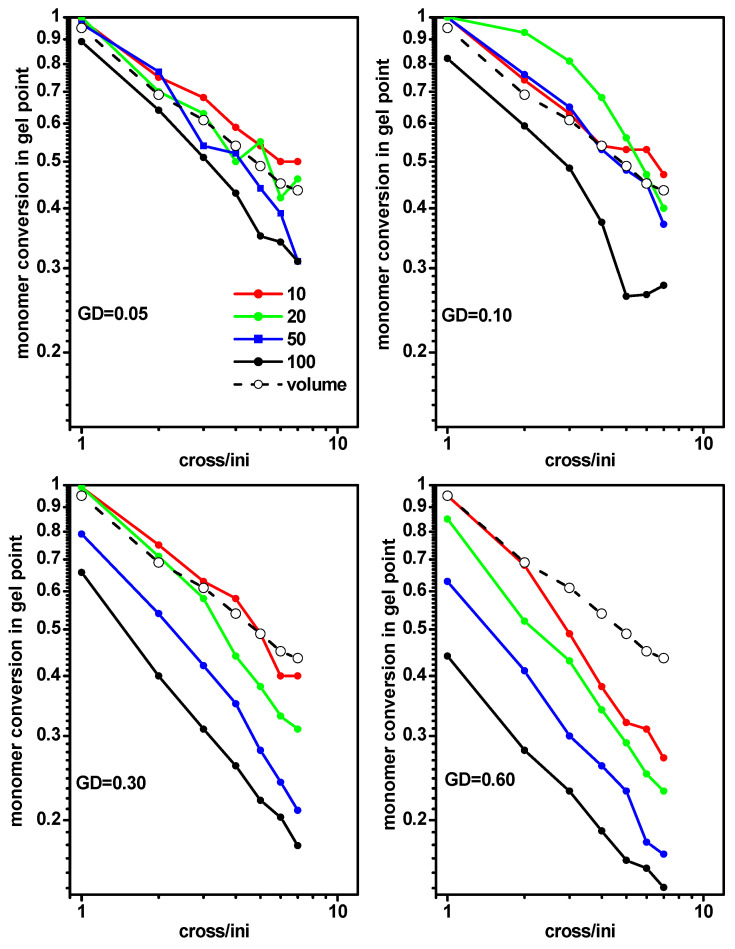
The conversion of monomer at the gel point for polymerization of grafted chains in a slit as a function of the ratio [Ini]/[X] for different grafting densities. The widths of the slit are given in the inset. The monomer conversion at gel point for a free volume are marked by the dashed line.

**Figure 11 ijms-24-02701-f011:**
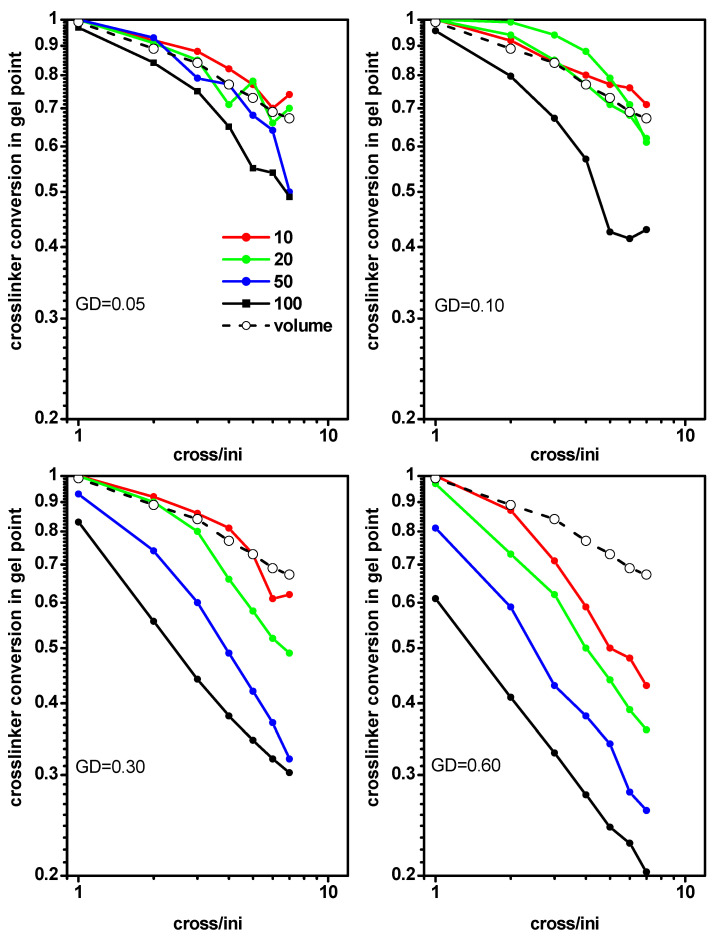
The conversion of cross-linker at the gel point for polymerization of grafted chains in a slit as a function of the ratio [Ini]/[X] for different grafting densities. The widths of the slit are given in the inset. The cross-linker conversion at gel point for a free volume are marked by the dashed line.

**Figure 12 ijms-24-02701-f012:**
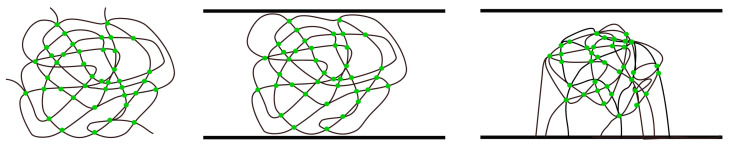
A scheme of the simulated systems: free chains in a solution (system A, **left**), free chains in a slit (system B, **middle**) and chains grafted to one wall of the slit (system C, **right**). Intra- and inter-polymer cross-links are marked as green points.

## Data Availability

The data that support the findings of this study are available from the corresponding author upon reasonable request.

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
