# Peer review of "The Influence of Constraints on Gelation in a Controlling/Living Copolymerization Process"

_ijms, 2023, doi:10.3390/ijms24032701_

Round 1

Reviewer 1 Report

In this work, Polanowski et al. apply a dynamic lattice liquid model to evaluate the gel point in living crosslinking copolymerization under confinement. The work is interesting but should be significantly polished before publication. Comments and suggestions are given below.

-        Kinetic Monte Carlo simulations regarding crosslinked/grafted/surface-initiated systems should be acknowledged.

-        Please present the considered reactions, reaction probabilities and all model parameters (perhaps in the Supporting Information). Please also mention for which monomer, crosslinker and initiator the considered parameters are representative of (also note that in the conclusion it is said that realistic reaction parameters are used). In general, a discussion on the considered DLL model parameters is advised.

-        The sentence starting on line 35 “However, it was shown that …” is unclear

-        Please check the definition of polymer brushes on line 41 (cf. polymer brushes in general)

-        Line 59: “solvent molecules were explicitly included in the model”: are it not bulk simulations (i.e. no solvent is present)?

-        Line 60: “cooperative Monte Carlo” should be briefly explained

-        Line 105: also a periodic boundary condition at z= 0 and z= 100 in case of “free chains in solution”?

-        It is mentioned that Figure 2 has a double logarithmic scale, but the x-axis seems to be on a linear scale. In any case, please comment on the choice for the logarithmic scale on the y-axis (cf. for a living polymerization with fast initiation and linear chains one expects a linear trend of the number average CL with conversion on a linear scale).

-        Something went wrong with the caption of Figure 3

-        The discussion of Figure 3 (left vs. right) should be improved and extended. Note that from a kinetic point of view a discussion in concentrations rather than in absolute number of molecules seems needed.

-        The discussion of Figure 7-12 needs to be improved. The applied format where each paragraph starts with “Figure X presents…” is not very appropriate.

-        Line 345: in view of the basic reaction scheme and constant reaction probabilities I strongly suggest to stress that the focus is on qualitative kinetic behavior only.

Many sentences need to be improved/corrected. Some examples are listed below:

-        Confinement, i.e. geometrical constraints changes the structure and physical properties of macromolecular systems [16]. Gelation process in confinement was studied experimentally showing the differences in structure and properties of such gel [17].

-        The main advantage of this model was the possibility of studies macromolecular systems at high densities

-        Line 305: “the influence of on the location of the gel point

-        Line 340: “We carrid”, “asimple

-        Line 342: “algorithmbased”, was employe

-        Line 345: “crossliniker”

Line 352: “even if they were of order of magnitude
